# Pre-Existing Comorbidities Diminish the Likelihood of Seropositivity after SARS-CoV-2 Vaccination

**DOI:** 10.3390/vaccines10081363

**Published:** 2022-08-20

**Authors:** Alok R. Amraotkar, Adrienne M. Bushau-Sprinkle, Rachel J. Keith, Krystal T. Hamorsky, Kenneth E. Palmer, Hong Gao, Shesh N. Rai, Aruni Bhatnagar

**Affiliations:** 1Division of Environmental Medicine, Christina Lee Brown Envirome Institute, Department of Medicine, University of Louisville School of Medicine, Louisville, KY 40202, USA; 2Center for Predictive Medicine for Biodefense and Emerging Infectious Diseases, Department of Medicine, University of Louisville School of Medicine, Louisville, KY 40202, USA; 3Center for Integrative Environmental Health Sciences, Department of Medicine and Department of Biostatistics and Bioinformatics, University of Louisville School of Medicine, Louisville, KY 40202, USA; 4Biostatistics and Bioinformatics Facility, James Graham Brown Cancer Centre, Department of Biostatistics and Bioinformatics, University of Louisville, Louisville, KY 40202, USA; 5Department of Bioinformatics and Biostatistics, University of Louisville, Louisville, KY 40202, USA

**Keywords:** SARS-CoV-2 antibody, antibody response, serology, serostatus, comorbidity

## Abstract

Background: The impact of chronic health conditions (CHCs) on serostatus post-severe acute respiratory syndrome coronavirus 2 (SARS-CoV-2) vaccination is unknown. Methods: We assessed serostatus post-SARS-CoV-2 vaccination among fully vaccinated adult residents of Jefferson County, Kentucky, USA, from April 2021 to August 2021. Serostatus was determined by qualitative analysis of SARS-CoV-2-specific Spike IgG antibodies via enzyme-linked immunoassay (ELISA) in peripheral blood samples. Results: Of the 5178 fully vaccinated participants, 51 were seronegative and 5127 were seropositive. Chronic kidney disease (CKD) and autoimmune disease showed the highest association with negative serostatus in fully vaccinated individuals. The absence of any CHC was strongly associated with positive serostatus. The risk of negative serostatus increased as the total number of pre-existing CHCs increased. Similarly, the use of two or more CHC-related medications was associated with seronegative status. Conclusions: The presence of any CHC, especially CKD or autoimmune disease, increased the likelihood of seronegative status among individuals who were fully vaccinated to SAR-CoV-2. This risk increased with a concurrent increase in number of comorbidities, especially with multiple medications. The absence of any CHC was protective and increased the likelihood of a positive serological response. These results will help develop appropriate guidelines for booster doses and targeted vaccination programs.

## 1. Introduction

The severe acute respiratory syndrome coronavirus 2 (SARS-CoV-2) and its mutated variants continue to increase morbidity and mortality worldwide [1]. Congruent with infection prevention and control measures, wide-spread vaccination is an effective means of containing and ending the Coronavirus Disease 2019 (COVID-19) pandemic [2]. However, in addition to the continued expansion of vaccination efforts, it is important to identify factors that influence vaccine efficacy, especially with recent reports of waning vaccine effectiveness after 6–8 months [3,4,5].

Effectiveness of the vaccines among non-immunocompromised individuals in the US is documented [6]. However, there are sparse data comparing vaccine efficacy between individuals with and without pre-existing comorbidities, and their relative effect on vaccination response. In vivo neutralization activity is a strong predictor of the protection from COVID-19 acquired from vaccination [7,8]. Although neutralization immunoassays are highly predictive of immune protection, they are severely limiting for large-scale deployment due to the technical complexities, low throughput, and a delayed turnaround time. [9]. Alternately, a high-throughput serological testing workflow is relatively easy to set up and can provide accurate and precise assessment of the presence or absence of SARS-CoV-2 Spike protein (IgG)-specific antibodies from blood samples. Our group has previously shown that serological tests can be used as a surrogate for immunological-response screening [10]. Gaining insights into factors which affect the serological response to vaccines will guide the deployment of boosters worldwide and help identify individuals who are at greater risk of breakthrough infections. Therefore, we sought to study the effect of comorbidities on serostatus post-SARS-CoV-2 vaccination.

## 2. Materials and Methods

### 2.1. Design and Study Population

Data for the study were collected under the Co-Immunity Project, which is a federally funded ongoing population-based study for the surveillance of SARS-CoV-2 in Jefferson County, Kentucky. The study was initiated in June 2020 and has completed 8 rounds of testing through October 2021. The SARS-CoV-2 vaccines were made widely available to the public in Kentucky starting April 2021. Data reported here are from April 2021 to August 2021. Participants were 18-years-and-older residents of Jefferson County who provided signed consent. This study and all the protocols were approved by the Institutional Review Board of the University of Louisville (IRB # 20.0393).

### 2.2. Vaccination Status

For analysis, only fully vaccinated participants were included. As per the Centers for Disease Control and Prevention (CDC), participants were considered “Fully Vaccinated” only if their final dose (2nd dose for Moderna or Pfizer-BioNTech and 1st dose for Janssen) was ≥14 days prior to the study appointment [11]. All participants in this study were within 9 months of their final vaccination dose.

### 2.3. Health History

Health history regarding pre-existing chronic health conditions and medications was self-reported by the study participants. History of lupus, rheumatoid arthritis, celiac disease, spondylitis, Graves’s disease, etc., was combined into a single variable, “Autoimmune Diseases”. Regular use of medications involved with immune-suppression, such as steroids, Secukinumab, Baricitinib, Tofacitinib, Methotrexate, etc., was combined into a single variable, “Immunosuppressants”. History of all modalities of cancer treatments, including chemotherapy, radiation therapy, surgery, immunotherapy, etc., was combined into a single variable, “Cancer Treatment”. Use of angiotensin converting enzyme inhibitors (ACEi) or angiotensin receptor blocker (ARB) was combined into a single variable, “ACEi or ARB”.

### 2.4. Human Samples and Serology

Trained staff collected nasopharyngeal swab and finger-prick blood samples. Samples were analyzed for infection by reverse-transcription polymerase chain reaction (RT-PCR) [12]. Serostatus was determined by measuring levels of SARS-CoV-2 Spike protein–specific immunoglobulin (Ig) G (Spike IgG) antibodies via enzyme-linked immunoassay (ELISA) in peripheral blood samples, as reported previously [10]. Serostatus is reported as a qualitative assessment (positive or negative) based on results from the ELISA test.

### 2.5. Statistical Analysis

Two study groups were defined *a priori*: seronegative and seropositive. The primary objective was to compare the univariate relationship of chronic health conditions (CHCs) between seronegative versus seropositive statuses. The secondary objective was to examine the adjusted associations between clinical characteristics and seronegative status. Participant characteristics are expressed as mean ± standard deviation for continuous variables, and frequency (%) for categorical variables. Relative magnitude of negative serostatus was estimated by odds ratio (OR) and 95% confidence intervals (CIs). 

For the secondary objective, multivariable logistic regression models were built to calculate the adjusted OR for negative serostatus in participants with CHCs or taking medications. Models were adjusted for age, sex, CHCs, and medications. The *p*-value was considered significant at <0.05. All statistical analyses were performed by using SAS, version 9.4 (SAS Institute, Inc., Cary, NC, USA). 

## 3. Results

### 3.1. Cohort

Of the 7046 participants enrolled from April 2021 to August 2021, a total of 1868 were excluded from the analysis because 802 were unvaccinated, 487 were missing vaccination dates or other vaccination related critical data, 247 were missing critical demographic information or medical history, and 332 were not “Fully Vaccinated”, as defined in Methods (Figure 1). The final study dataset included 5178 participants, of which 51 were seronegative and 5127 were seropositive. The general cohort information is presented in Table 1. Briefly, the mean age was higher in the seronegative group (69 ± 25 years), as compared to the seropositive group (62 ± 23 years; *p* = 0.024). There was no difference in sex, race, or tobacco use between the two study groups. 

### 3.2. CHCs and Serostatus 

Among those with no CHC, 21.6% were seronegative, whereas 42.4% were seropositive (OR = 0.37, 95% CI: 0.19–0.073; *p* = 0.003) (Table 1). Among those with diabetes, 17.7% were seronegative, whereas 10.6% were seropositive (OR = 3.27, 95% CI: 1.35–7.94; *p* = 0.01). Among those with hypertension, 45.1% were seronegative, whereas 33.6% were seropositive (OR = 2.9, 95% CI: 1.38–6.11; *p* = 0.003). Among those with heart disease, 13.7% were seronegative, whereas 7.6% were seropositive (OR = 3.56; 95% CI: 1.37–9.23; *p* = 0.013). For autoimmune diseases, 31.4% were seronegative, whereas 5.4% were seropositive (OR = 11.34, 95% CI: 5.21–24.69; *p* < 0.0001). Among those with chronic kidney disease (CKD), 11.8% were seronegative, whereas 1.7% were seropositive (OR = 13.49, 95% CI: 4.88–37.3; *p* < 0.0001). 

Individuals with a composite of any cardiovascular disease (CVD) were 49% seronegative, whereas 36.2% were seropositive (OR = 2.93, 95% CI: 1.4–6.11; *p* = 0.003). Similarly, among individuals with a composite of any CHC, 78.4% were seronegative, whereas 57.6% were seropositive (OR = 2.68, 95% CI: 1.37–5.24; *p* = 0.003).

### 3.3. Medications and Serostatus

A history of immunosuppressants (OR = 23, 95% CI: 12.1–43.25; *p* < 0.0001) and cancer treatments (OR = 12, 95% CI: 4.17–36.88; *p* = 0.001) was also higher in the seronegative group. A history of ACEi or ARB was not associated with serostatus (OR = 0.82; 95% CI: 0.34–1.99; *p* = 0.66).

### 3.4. Adjusted Associations between Clinical Characteristics and Seronegative Status

Age ≥ 65 years, sex, and reported history of tobacco product use were not significantly associated with seronegative status (Table 2). However, the presence of even one reported CHC was significantly associated with seronegative status (OR = 2.69; 95% CI: 1.25–5.79). Similarly, the association between pre-existing CHCs and seronegative status strengthened with an increase in the number of CHCs from two (OR = 2.82; 95% CI: 1.14–7.0) to three or more (OR = 4.52; 95% CI: 1.68–12.14). 

## 4. Discussion 

In this study of 5178 fully vaccinated participants enrolled during the beginning of the Delta variant (B.1.617.2 and AY lineages) spread, we found that those reporting no CHCs were more likely to develop detectable antibody levels (i.e., seropositivity) after completing vaccination regimen. Conversely, those reporting a pre-existing comorbid condition were less likely to develop a seropositive status after immunization. The highest risk of not developing a detectable antibody response was seen with CKD; however, those reporting CVD, diabetes, or hypertension were also less likely to develop seropositive status after vaccination. The odds of a seronegative status increased with an increase in the number of comorbidities. 

More than 10.6 billion COVID-19 vaccine doses have been administered worldwide, and most of the “fully vaccinated” population worldwide completed their final dose at least 6–8 months ago [1]. Similarly, most of the adults in the United States who received their first booster dose completed the booster regimen at least 11 months ago [13]. There are growing concerns about waning vaccine effectiveness, including the first booster dose, after 6–8-months [3,4,5]. Therefore, it is important to identify factors that may play a role in influencing the vaccine effectiveness. Although it is widely suspected that pre-existing comorbidities play a critical role in determining responses to immunization [14], few data are available to determine the role of comorbidities on serological response or vaccine efficacy in the public. We found that <1% of our study population failed to develop an appreciable response to the vaccine despite the high rates of diabetes (10.7%), hypertension (33.7%), and cancer (6.8%) in our cohort that are comparable to national rates of these comorbidities. Neutralization immunoassays are the gold standard to measure the effect of immune protection post-vaccination, but they have severe logistical limitations for large-scale use, including maintaining a Biosafety Level 3 facility [9]. On the other hand, serological assays can identify the presence or absence of SARS-CoV-2-specific antibodies in blood samples in a high-throughput setting. Even though positive serology does not ensure the competence of an immune response, there is a significant relationship between a seropositive status and SARS-CoV-2-neutralization potential via microneutralization and Plaque Reduction Neutralization Assays (PNAs) [10]. This suggests that those who fail to establish a measurable seropositive status are more likely to remain vulnerable to SARS-CoV-2 infection. Furthermore, even though a seropositive status does not guarantee a successful vaccination effect (i.e., SARS-CoV-2 neutralization), a seronegative status excludes any possibility of virus neutralization [15]. 

About 52% adults in the United States have at least one CHC, and about 27% have multiple CHCs [16]. According to the data presented in our study, almost half of the adult population in the United States has a two-fold chance of being vulnerable to SARS-CoV-2 (or re-infection), even if they are fully vaccinated. The top risk factors for CKD, such as diabetes, hypertension, and CVD [17], are also the most commonly implicated risk factors of severe COVID-19 outcomes [18]. The results from our study suggest that pre-existing CKD might be a significant threat to SARS-CoV-2 vaccine efficacy, even among fully vaccinated individuals, as has been suspected previously [19,20]. The high likelihood of negative serostatus among individuals with CKD could be a synergistic influence of these CHCs in exacerbating an underlying immune dysfunction, which is well-known in CKD [21,22]. Diminished innate immune response due to insufficient activation of antigen-presenting cells [23] or tandem impairment of innate and adaptive immune responses is commonly observed in the presence of CKD [22,24]. Similarly, impaired immune response is the hallmark of autoimmune disease [25]. The etiological origins of autoimmune diseases differ from CKD or COVID-19 severity, which is due to specific underlying substrata (i.e., CHCs) in CKD or COVID-19 severity. However, the immune dysfunction stemming from maladaptation of innate and adaptive immune responses is a common occurrence between autoimmune diseases and CKD [21,26]. The involvement of dysregulated and abnormal signaling of the toll-like receptor in autoimmune diseases [27,28], as well as CKD [29], is a prime example of this interrelationship. Finally, the immunosuppressive effects of various modalities used to treat autoimmune disease or CKD may also play an important role in suppressing an appreciable immune response post-SARS-CoV-2 vaccination, resulting in a seronegative status. Overall, our findings suggest that the presence of these comorbidities, either individually or in concurrence, significantly increases the likelihood of SARS-CoV-2 vaccine failure.

The Global Burden of Disease Study (GBD) has estimated one in five individuals worldwide to be at risk for severe COVID-19, primarily due to their burden of pre-existing CHCs [19]. There is a significant overlap in diseases reported by the GBD and those found to increase the likelihood of a seronegative status in our study [30]. Cancer, a major CHC in GBD, showed a very modest association with seronegative status in our study. However, current treatment for cancer was strongly associated with seronegative status, indicating that treatment status might be a more accurate clinical indicator of impaired immune response rather than a clinical diagnosis of cancer. Some studies have shown a significant association between ACEi or ARB use and risk of SARS-CoV-2 infection and severity of health outcomes [31,32]. In contrast to that, our study did not find any association, positive or negative, between ACEi or ARB use and serostatus. This suggests that the use of ACEi or ARB may not affect vaccine efficacy, even if their use has the potential to influence the risk of disease or its severity. 

Additional work is required to elucidate whether the presence of these comorbidities and other risk factors, such as smoking, influence the effectiveness of the virus neutralization capabilities post-vaccination. The results of such studies will help better shape the guidelines for booster doses and future vaccination programs. 

## 5. Limitations

Limitations of the study include a lack of SARS-CoV-2 titer or neutralization data, which were not the primary objective of the study but can be used to evaluate a continuous relationship between co-morbid conditions and antibody titer and neutralization efficacy. Serological assessment is not a definitive method to measure antibody efficacy, but seropositive status has a strong correlation with virus-neutralization potential. The main goal of this study was to identify differences in serostatus among fully vaccinated individuals, not to study the difference between individual vaccines. Finally, this study is cross-sectional in design, so it does not allow us to report changes over time.

## 6. Conclusions

Among fully vaccinated individuals, the presence of any CHC, especially CKD or autoimmune disease, increased the likelihood of seronegative status by multi-fold. This risk increased with a concurrent increase in the number of comorbidities. CHCs requiring long-term treatment, especially immunosuppressants or cancer treatments, increased the likelihood of a seronegative status. Absence of any CHC was protective and increased the likelihood of a positive serological response post-vaccination. These results will help develop appropriate guidelines for booster doses and targeted vaccination programs.

## Figures and Tables

**Figure 1 vaccines-10-01363-f001:**
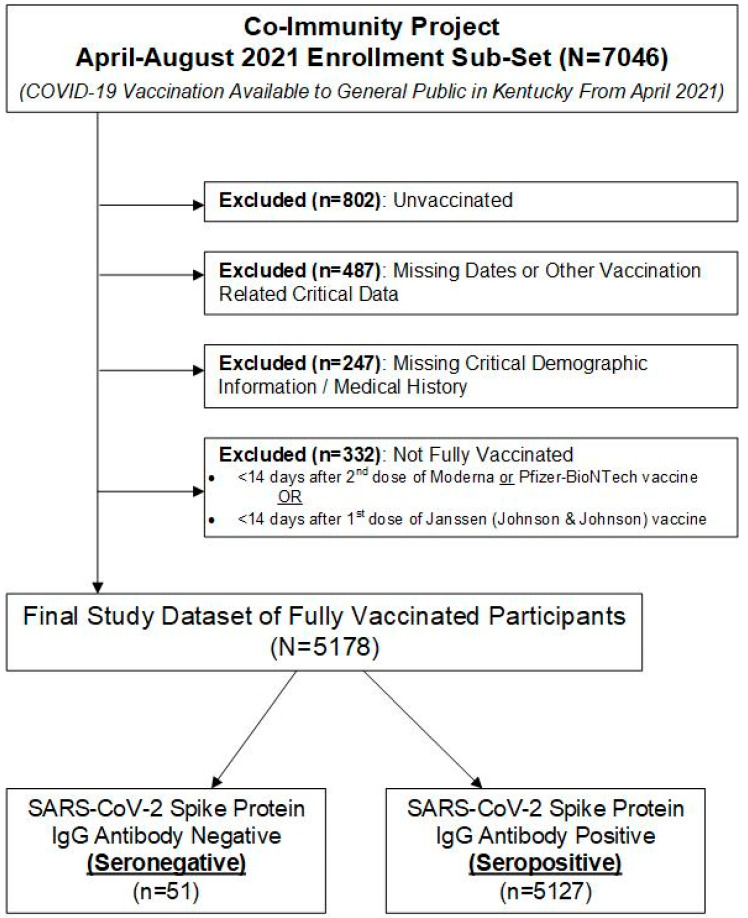
CONSORT flow diagram of dataset identification.

**Table 1 vaccines-10-01363-t001:** Univariate comparison of clinical characteristics between study groups.

Characteristics	Antibody Negative * (*n* = 51)	Antibody Positive * (*n* = 5127)	Odds Ratio (95% CI)	*p*-Value
Age, Median ± IQR	69 ± 25	62 ± 23		**0.024**
Sex, Female N (%)	37 (72.6)	3393 (66.2)		0.34
Race, White N (%)	48 (94.1)	4457 (86.9)	2.41 (0.75–7.74)	0.13
Any Tobacco Use	2 (3.9)	201 (3.9)	1 (0.24–4.14)	1.00
Chronic Health Conditions (CHCs)				
None, N (%)	11 (21.6)	2176 (42.4)	0.37 (0.19–0.73)	**0.003**
Diabetes, N (%)	9 (17.7)	544 (10.6)	3.27 (1.35–7.94)	**0.010**
Hypertension, N (%)	23 (45.1)	1723 (33.6)	2.9 (1.38–6.11)	**0.003**
Heart Disease, N (%)	7 (13.7)	389 (7.6)	3.56 (1.37–9.23)	**0.013**
Autoimmune Disease, N (%)	16 (31.4)	279 (5.4)	11.34 (5.21–24.69)	**<0.0001**
Cancer, N (%)	5 (9.8)	347 (6.8)	2.85 (0.98–8.25)	0.06
Thyroid Disease, N (%)	6 (11.8)	547 (10.7)	2.17 (0.8–5.89)	0.13
Chronic Kidney Disease, N (%)	6 (11.8)	88 (1.7)	13.49 (4.88–37.3)	**<0.0001**
Composite CHCs				
Cardiovascular Disease, N (%)	25 (49.0)	1856 (36.2)	2.93 (1.4–6.11)	**0.003**
Any CHC, N (%)	40 (78.4)	2951 (57.6)	2.68 (1.37–5.24)	**0.003**
Medications				
None, N (%)	27 (52.94)	3852 (75.1)	0.37 (0.21–0.65)	**0.0003**
All Medications, N (%)	24 (47.06)	1275 (24.9)	2.69 (1.54–4.67)	**0.0003**
ACEI or ARB, N (%)	6 (11.76)	1041 (20.3)	0.82 (0.34–1.99)	0.66
Immunosuppressants, N (%)	17 (33.33)	106 (2.1)	22.88 (12.1–43.25)	**<0.0001**
Cancer Treatments, N (%)	4 (7.84)	46 (0.9)	12.41 (4.17–36.88)	**0.001**

Abbreviations: ACEI—angiotensin converting enzyme inhibitor; ARB—angiotensin receptor blocker; CHC—chronic health conditions; IQR—interquartile range. * Serostatus determined by presence/absence of SARS-CoV-2 Spike Protein (IgG) antibodies in peripheral blood via ELISA. Statistically significant *p-values* are represented in bold font.

**Table 2 vaccines-10-01363-t002:** Multivariable association between clinical characteristics and SARS-CoV-2 Spike IgG antibody negative status.

Characteristics	Referent Group	Odds Ratio	95% CI
Age ≥ 65 years	<65 Years	1.13	0.64–1.98
Sex, Female (%)	Male	1.4	0.77–2.54
Composite Chronic Health Conditions (CHCs)			
1 CHC	No CHC	**2.69**	**1.25–5.79**
2 CHCs	No CHC	**2.82**	**1.14–7.0**
≥3 CHCs	No CHC	**4.52**	**1.68–12.14**
Composite Medications			
1 medication	No Medications	1.43	0.76–2.71
≥2 medications	No Medications	**6.08**	**2.01–18.35**

Abbreviations: CHC—chronic health conditions; CI—confidence interval. Statistically significant *p-values* are represented in bold font.

## Data Availability

Primary de-identified data supporting the findings of this study are available through the corresponding author (Aruni Bhatnagar: Aruni.bhatnagar@louisville.edu) upon request.

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
