# Peer review of "Pre-Existing Comorbidities Diminish the Likelihood of Seropositivity after SARS-CoV-2 Vaccination"

_vaccines, 2022, doi:10.3390/vaccines10081363_

Round 1
Reviewer 1 Report
Only some minor comments
1. Introduction is section 1, then "MATERIALS AND METHODS" should be section 2, followed by the rest sections.
2. Check the format of Table 2.
3. In line 211 "Interestingly, our study", the sentence is not finished.
4. In Discussion, the authors indicated that "Although diabetes, hypertension, and CVD are well-known risk factors for severe COVID-19 outcomes,(17) our study shows that CKD is the biggest threat, even among fully vaccinated individuals as has been suspected previously.(18, 19)"
"Although" in here somehow implies that diabetes, hypertension, and CVD may not be important risk factors (as CKD). If so, the authors should provide further discussion. However, based on the results presented in this study, diabetes, hypertension, and CVD are risk factors, and the authors found CKD is the biggest one.
Author Response
REVIEWER # 1 Comments and Suggestions for Authors
Only some minor comments
- Introduction is section 1, then "MATERIALS AND METHODS" should be section 2, followed by the rest sections.
We would like to thank the reviewer for bringing up this important issue. After re-checking our submission documents, we have found that there was a formatting error on the submission software’s side. The sections in our submission were originally labeled as described by the reviewer but were formatted incorrectly post-submission. We have made the necessary updates to the revised (R1) submission as requested by the reviewer.
- Check the format of Table 2.
We thank the reviewer for bringing up this important formatting issue. We have updated the format for Table 2 as per the manuscript template provided by the journal editor.
- In line 211 "Interestingly, our study", the sentence is not finished.
We thank the reviewer for bringing up this very important error. We have deleted the sentence “Interestingly, our study”, since that is already addressed in a previous sentence “In contrast to that, our study did not find any association, positive or negative, be-tween ACEi or ARB use and serostatus”.
- In Discussion, the authors indicated that "Although diabetes, hypertension, and CVD are well-known risk factors for severe COVID-19 outcomes,(17) our study shows that CKD is the biggest threat, even among fully vaccinated individuals as has been suspected previously.(18, 19)"
"Although" in here somehow implies that diabetes, hypertension, and CVD may not be important risk factors (as CKD). If so, the authors should provide further discussion. However, based on the results presented in this study, diabetes, hypertension, and CVD are risk factors, and the authors found CKD is the biggest one.
We thank the reviewer for noticing this important grammatical issue. We have amended the sentence as below:
“The top risk factors for CKD like diabetes, hypertension, and CVD,(17) are also the most commonly implicated risk factors of severe COVID-19 outcomes.(18) Results from our study suggest that pre-existing CKD might be a significant threat to SARS-CoV-2 vaccine efficacy, even among fully vaccinated individuals, as has been suspected previously.(19, 20)”

Reviewer 2 Report
Congratulations to the authors, the work is well structured, scientifically valid and interesting. The study is relevant given that it shows how in subjects with co-morbidities and chronic diseases the effectiveness of the two doses of the vaccine is reduced. I have only a few minor considerations.
Minor questions:
1) In the description of the data I do not see the measured antibody levels (anti-spike IgG antibodies) indicated. I assume that the ELISA test done to measure the levels of antibodies specific to the spike returned a quantitative titer (U / ml), this should be indicated, also indicating the positive / negative cut-off in the materials and methods section. Or did the ELISA test indicate only positive or negative for the presence of anti-spike antibodies? this should be indicated in the methods section.
2) If the test gave a quantitative titer, in my opinion the average level of the antibody titer should be indicated for the different categories. Subjects with seroconversion and subjects without seroconversion (entering the data in the results section).
3) With reference to subjects with no seroconversion (after vaccination) who were mainly subjects with CKD or autoimmune diseases, the authors could insert some speculation on the mechanisms that according to literature data in these subjects reduce the response to the vaccine, indicating any similarities between subjects with autoimmune diseases and CKD. It only takes two or three sentences in the discussion section.
Author Response
REVIEWER # 2 Comments and Suggestions for Authors
Congratulations to the authors, the work is well structured, scientifically valid and interesting. The study is relevant given that it shows how in subjects with co-morbidities and chronic diseases the effectiveness of the two doses of the vaccine is reduced. I have only a few minor considerations.
Minor questions:
- In the description of the data I do not see the measured antibody levels (anti-spike IgG antibodies) indicated. I assume that the ELISA test done to measure the levels of antibodies specific to the spike returned a quantitative titer (U / ml), this should be indicated, also indicating the positive / negative cut-off in the materials and methods section. Or did the ELISA test indicate only positive or negative for the presence of anti-spike antibodies? this should be indicated in the methods section.
We thank the reviewer for bringing up this key question. We did not report antibody levels in this manuscript. As described in the “Human Samples and Serology” sub-section of the “Materials and Methods” section, serological status was determined based on qualitative assessment of positive vs negative anti-spike IgG antibodies. We have amended the sub-section to better illustrate the antibody measurement as described below:
“Trained staff collected nasopharyngeal swab and finger prick blood samples. Samples were analyzed for infection by reverse transcription polymerase chain reaction (RT-PCR).(12) Serostatus was determined by measuring levels of SARS-CoV-2 Spike protein specific immunoglobulin (Ig) G (Spike IgG) antibodies via enzyme-linked immunoassay (ELISA) in peripheral blood samples as reported previously.(10) Serostatus is reported as a qualitative assessment (positive or negative) based on results from the ELISA test described above.”
- If the test gave a quantitative titer, in my opinion the average level of the antibody titer should be indicated for the different categories. Subjects with seroconversion and subjects without seroconversion (entering the data in the results section).
We thank the reviewer for bringing up this key question, however we have not reported quantitative titers in this manuscript. As clarified in Question # 1 above, only qualitative (positive vs negative) assessments of SARS-CoV-2 specific Spike Protein antibodies (IgG) were analyzed in this study.
- With reference to subjects with no seroconversion (after vaccination) who were mainly subjects with CKD or autoimmune diseases, the authors could insert some speculation on the mechanisms that according to literature data in these subjects reduce the response to the vaccine, indicating any similarities between subjects with autoimmune diseases and CKD. It only takes two or three sentences in the discussion section.
We thank the reviewer for bringing to our attention this important point. We have amended the “Discussion” section as described below to address this important concern:
“The top risk factors for CKD like diabetes, hypertension, and CVD,(17) are also the most commonly implicated risk factors of severe COVID-19 outcomes.(18) Results from our study suggest that pre-existing CKD might be a significant threat to SARS-CoV-2 vaccine efficacy, even among fully vaccinated individuals, as has been suspected previously.(19, 20) The high likelihood of negative serostatus among individuals with CKD could be a synergistic influence of these CHCs in exacerbating an underlying immune dysfunction, which is well-known in CKD.(21, 22) Diminished innate immune response due to insufficient activation of antigen presenting cells,(23) or tandem impairment of innate and adaptive immune responses are commonly observed in the presence of CKD.(22, 24) Similarly, impaired immune response is the hallmark of autoimmune disease.(25) Etiological origins of autoimmune diseases differ from CKD or COVID-19 severity, which is due to a specific underlying substrata (i.e., CHCs) in CKD or COVID-19 severity. However, the immune dysfunction stemming from maladaptation of innate and adaptive immune responses is a common occurrence between autoimmune diseases and CKD.(21, 26) The involvement of dysregulated and abnormal signaling of the toll-like receptor in autoimmune diseases (27, 28) as well as CKD (29) is a prime example of this interrelationship. Finally, the immunosuppressive effects of various modalities used to treat autoimmune disease or CKD may also play an important role in suppressing an appreciable immune response post SARS-CoV-2 vaccination, resulting in a seronegative status. Overall, our findings suggest that presence of these comorbidities, either individually or in concurrence, significantly increases the likelihood of SARS-CoV-2 vaccine failure”
